# Low thoracic skeletal mass index, a novel marker to predict recurrence of aspiration pneumonia in the elderly stroke patients

**Bo Mi Gil[1], Sun Im[2], Yu Jin Hong[3], Hye Seon Kang[4]***

**1** Department of Radiology, Bucheon St. Mary's Hospital, College of Medicine, The Catholic University of Korea, Seoul, Republic of Korea, **2** Department of Rehabilitation Medicine, Bucheon St. Mary's Hospital, College of Medicine, The Catholic University of Korea, Seoul, Republic of Korea, **3** Division of Pulmonary, Allergy and Critical Care Medicine, Department of Internal Medicine, Incheon St. Mary's Hospital, College of Medicine, The Catholic University of Korea, Seoul, Republic of Korea, **4** Division of Pulmonary, Allergy and Critical Care Medicine, Department of Internal Medicine, Bucheon St. Mary's Hospital, College of Medicine, The Catholic University of Korea, Seoul, Republic of Korea

☯ These authors contributed equally to this work.
* beyer_kr@catholic.ac.kr

**Data Availability Statement:** All relevant data are within the manuscrtip and its Supporting information files.

## Abstract

### Purpose

We investigated whether thoracic skeletal muscle mass index at the diagnosis of aspiration pneumonia (AP) is a predictor for AP recurrence and explored predicting factors for AP recurrence in patients with stroke.

### Patients and methods

This study retrospectively reviewed data of patients with AP who were diagnosed with stroke and who had full medical follow-up data from January 2014 to July 2020 in the Catholic University of Korea Bucheon St. Mary's Hospital. AP was defined based on clinical signs and/or symptoms suggestive of pneumonia and radiologic findings of pneumonic infiltrations in the dependent portions of the lung. We measured thoracic muscle volume using the cross-sectional area (CSA) of the erector spinae muscle (ESMCSA, cm$^2$) at the 12th vertebral region. Computed tomography scans at the time of AP diagnosis during the acute stroke period were used for analysis and respective CSAs were divided by height squared (m$^2$) to yield the muscle index at T12 (T12MI, cm$^2$/m$^2$) to normalize for stature. Multivariate logistic regression models were used to investigate relationships between clinical parameters and AP recurrence.

### Results

During the study period, a total of 268 stroke patients with dysphagia who developed AP were analyzed. The mean T12MI of patients with and without recurrence of AP was 622.3 ±184.1 cm$^2$/m$^2$ and 708.1±229.9 cm$^2$/m$^2$, respectively ($P = 0.001$). Multivariate logistic regression revealed that lower T12MI ($P = 0.038$) and older age ($P = 0.007$) were independent predictors of AP recurrence in patients with stroke and dysphagia.

**Funding:** This work was supported by a National Research Foundation of Korea (NRF) grant funded by the Korean government (no. 2020R1F1A1065814) and Technology Innovation Program (No. 20018182) funded by the Ministry of Trade, Industry & Energy (MOTIE, Korea). The funders had no role in study design, data collection and analysis, decision to publish, or preparation of the manuscript.

**Competing interests:** The authors have declared that no competing interests exist.

## Conclusion

Low thoracic muscle index at the diagnosis of initial AP after stroke can predict subsequence AP recurrence.

## Introduction

Stroke is the second leading cause of death in the world, and its prevalence is projected globally [1]. Dysphagia is a common complication of stroke and a critical factor for aspiration pneumonia (AP) development and mortality in patients with stroke [2]. AP is caused by the aspiration of bacteria from the oropharyngeal cavity or from gastrointestinal fluid into the lung [3]. In a prospective observational study, AP accounts for 31.4% of hospitalized community acquired pneumonia or health care associated pneumonia. Approximately 80% of pneumonia cases that require hospitalization are associated with aspiration in elderly patients aged 80 years or older [4]. In meta-analysis including 19 studies, AP increased in-hospital mortality (relative risk, 3.62) and 30-day mortality (relative risk, 3.57) [5]. Established risk factors for recurrence of AP include dementia, Eastern Cooperative Oncology Group-Performance Status 4, taking sleeping medications, and laryngeal paralysis [6, 7]. As AP is primarily caused by impairment of swallowing and the cough reflex, other clinical factors that can predict functional decline are needed.

Sarcopenia is defined as loss of muscle mass, physical performance, and strength [8]. Sarcopenia involves the accelerated loss of muscle mass and function that is related to increased adverse clinical outcomes including falls, functional decline, frailty, and mortality. Sarcopenia was first described in 1988 and its clinical significance was sufficiently recognized to be registered in the International Classification of Diseases-10 code in 2016 [9]. The swallowing muscles have a different embryological origin from somatic muscles, and they receive continuous input stimulus from the respiratory system. Also the swallowing muscles are inevitably affected by malnutrition and disuse [10]. A few studies have shown the relationship between sarcopenia and pneumonia in older people including the onset of pneumonia development or 90-day mortality according to sarcopenia [11, 12].

After discussion of the relationship between malnutrition and dysphagia in 1992, the term "sarcopenic dysphagia" was first used by Kuroda et al. in 2012 [13, 14]. In patients with acute stroke, age-related muscle atrophy was associated with swallowing disorders post- stroke apart from features being determined by the acute stroke itself such as site and size of stroke [15]. A decrease in the mass or strength of the swallowing muscles is associated with swallowing dysfunction [16]. In patients with stroke, patients with dysphagia are three times, and those with confirmed aspiration eleven times, more likely to develop pneumonia [17, 18]. Thus, the identification of risk factors for AP development and prevention of AP recurrence is needed. Recurrence can result in prolonged hospital stays and lead to poor functional outcomes. While there have been numerous publications focusing on identifying individuals at risk of AP during acute stroke care, there remains limited information concerning which among these critical patients are particularly susceptible to subsequent infections.

Recent studies have advocated the use of the thoracic muscle volume using the CSA of the erector spinae muscle (ESM) [19]. However, the investigation of the relationship between low thoracic muscle volume and dysphagia-related AP in patients with stroke have not been explored.

This study aimed to investigate whether the presence of muscle mass index, as measured by the thoracic muscle volume at the diagnosis of AP is a predictor for the recurrence of future

AP and to explore predicting factors for the recurrence of AP in patients with stroke and dysphagia.

## Material and methods

### Participants

For this study, we conducted a retrospective review of data for patients diagnosed with dysphagia disorder due to acute and subacute stroke. The data collection period spanned from March 2014 to July 2020 and took place at a university-affiliated medical center with specialized acute stroke care settings. The data were accessed on 31 December 2022. The data was coded and used with a unique identification code for each subject. In 2,819 patients confirmed with dysphagia, 586 patients developed AP during the acute and subacute stroke period (within the first 4 weeks after stroke onset). Of these, 92 patients without medical records for follow-up and 229 patients who had no chest computed tomography (CT) were excluded. Finally, 268 patients were included in the analysis. The study flow is summarized in Fig 1.

This study was approved by the Clinical Research Ethics Committee of the Catholic Medical Center (approval number: HC22RISI0034). Researchers were permitted to conduct this study by accessing a dataset newly assigned with a serial number with personal information removed after ethical approval. In view of the retrospective nature of the study, the need for informed consent from patients was waived.

In accordance to previous past studies, the definition of AP was (1) the presence of respiratory symptoms, such as purulent sputum, tachypnea, and rales; (2) leukocytosis; (3) elevated temperature; (4) gravitational segment infiltration on a chest X-ray confirmed by both the radiologist and pulmonologist [20].

### Data

The following data were extracted from the patients' medical records: patient demographics, comorbidities, smoking status, skeletal muscle index using chest CT; obtained at the time of first AP during the acute stroke period; laboratory data, respiratory muscle function including

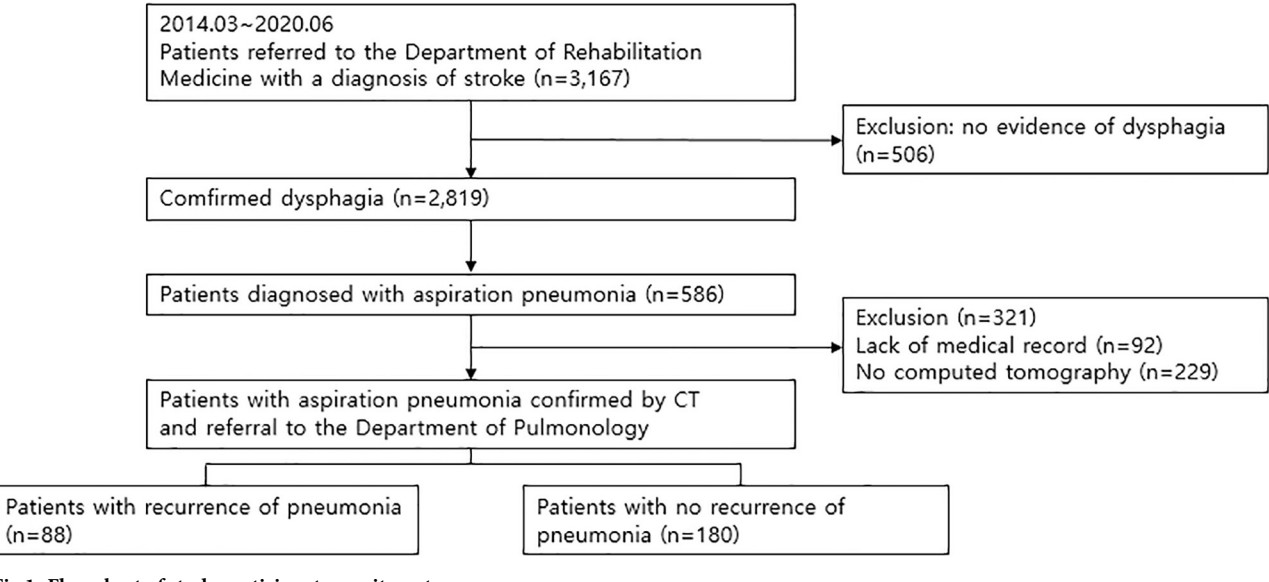

**Fig 1. Flow chart of study participant recruitment.**

the voluntary peak cough flow (PCF), dysphagia severity, functional disability obtained during the initial swallowing screening; survival status, and the dates of AP recurrence. Blood samples drawn within three days of AP onset were used.

## Respiratory pressure parameters

Electronic chart records of the voluntary PCF, maximal expiratory pressures (MEP) and maximal inspiratory pressures (MIP) were retrieved: For the former, the individuals were asked to perform a quick, short, and explosive cough on the peak flow meter, in compliance with the American Thoracic Society/European Respiratory Society standards [21]. The values were presented as the average values from five attempts. MIP and MEP were measured using a respiratory pressure meter (Micro-Plus Spirometer; Carefusion, Corp., San Diego, CA, USA) with a standard flange mouthpiece. In patients with stroke, severe facial palsy and unable to perfect lip seal, the therapist aided by holding the lips around the mouthpiece to minimize air leak. A live demonstration was performed by the physiotherapist and the patient was allowed to practice before the formal assessment [22].

## Dysphagia severity

Assessments performed within 2 weeks of the first AP were retrieved. Instrumental swallowing assessments were performed either through the Fiberoptic Endoscopic Evaluation of Swallowing or the videofluoroscopic swallowing study by an expert with more than 10 years' experience. The degree of swallowing impairment was recorded using the Functional Oral Intake Scale (FOIS) [23], an ordinal scale with level 1 indicating total tube feeding and worst level of swallowing ability and level 7 indicating full oral intake without diet modification. The severity and depth of aspiration assessed during the instrumental assessment was recorded with the penetration aspiration scale (PAS), which is an ordinal measure of aspiration severity and airway compromise (where 8 is severe aspiration past glottis with absence of cough response and 1 is no aspiration or penetration) [24]. In addition swallowing function was evaluated using screening tools including the Gugging Swallowing Screen (GUSS) [25] and the Mann Assessment of Swallowing Ability (MASA) [26], with lower scores indicating a more severe state.

## Functional disability

Levels of functional disability after brain lesions were determined using the modified Barthel index [27], Berg scales [28], the Mini-Mental State Examination (MMSE) [29]. These values, recorded within two weeks of AP were retrieved through the electronic charts.

## CT image analysis

All non-enhanced CT images at the diagnosis of AP were obtained during full inspiration in the supine position using a multidetector CT scanner with 64 channels (SOMATOM Sensation 64, Siemens, Germany). The detailed CT parameters were tube voltage, 120 kVp; tube current standard dosage (reference mAs, 60–120) with automatic exposure control; and slice thickness, 1.5–2.5 mm. The CT images were reviewed with mediastinal window settings (width, 450 Hounsfield units [HU]; level, 60 HU) by one thoracic radiologist with five years' experience, as a faculty chest radiologist. The ESMs' CSA at the level of the 12th thoracic vertebra was measured in all patients. The established muscle density is 40–100 HU, derived from the mean attenuation within the region of interest [30]. Based on this, the quantitative value of pure muscles' CSA was determined, excluding fat or calcification within the muscles, using a semi-automated system. The CSA of the paraspinal muscle ($cm^2$) at the T12 level was divided

by height squared ($m^2$) to obtain the skeletal muscle index (T12MI, $cm^2/m^2$), normalized for stature [31–35].

## Statistical analysis

The patient baseline demographics and clinical outcomes were compared in patients with and without AP recurrence. Pearson's chi-squared test was used to compare the discrete variables and the Student's $t$-test, standardized mean difference or analysis of variance to compare the continuous variables. Univariate analysis and multivariable logistic regression analysis were performed to identify the variables that independently increased the risk of AP. Hazard ratios (HRs) and the corresponding 95% confidence intervals (CIs) were calculated for the predictors that were significant in multivariate Cox regression analysis. A two-sided $p$-Value of $<0.05$ was considered statistically significant. All statistical analyses were performed using SPSS for Windows software (ver. 20.0; IBM Corp., Armonk, NY, USA).

## Results

During the study period, a total of 268 patients with stroke, dysphagia disorder and who had developed AP were analyzed. Of these, AP recurred in 32.8% of all subjects. Patients with AP were divided into two groups—those with and without recurrence. Mean time to recurrence was 152.5 days. The mean age (68.26 ± 13.31 vs. 73.68 ± 10.87, $P < 0.001$) was higher, but the mean value of body mass index (BMI) (21.27 ± 3.54 vs. 20.20 ± 2.93, $P = 0.014$), T12MI (708.1 ± 229.9 vs. 622.3 ± 184.1, $P = 0.001$), min PCF (84.6 ± 91.6 vs. 57.8 ± 59.2, $P = 0.004$), average PCF (100.0 ± 112.0 vs. 70.3 ± 66.8, $P = 0.007$) was significantly lower in the recurrence group. The proportion of those with successful nasogastric tube weaning (40.6% vs. 25.0%), $P = 0.018$) was lower in the recurrence group (Table 1). A comparison of paraspinal thoracic muscles at the T12 level between patients with and without AP is shown in Fig 2.

There were no significant differences in laboratory findings, including complete blood cell counts and inflammatory markers such as neutrophil lymphocyte ratio (NLR), platelet lymphocyte ratio (PLR) and C-reactive protein (CRP), between the two groups (Table 2).

In the correlation between T12MI and clinical parameters, and all the respiratory pressure meter values showed a significant correlation with T12MI, with the correlation coefficients being albumin (r = 0.2954, $P < 0.001$), protein (r = 0.1555, $P < 0.05$), hemoglobin (r = 0.2798, $P < 0.001$) and all the respiratory pressure meter values (Table 3). By contrast, CRP showed a significant negative correlation with T12MI, (r = -0.1238 ($P = 0.043$).

The results showed that peak cough flow (PCF) was the only respiratory parameter with a significant difference between the recurrence and non-recurrence groups. In contrast, other dysphagia parameters (FOIS, PAS, GUSS, MASA) and functional disability markers (Barthel index, Berg scales, MMSE) showed no significant differences, leading to their exclusion from the univariate and multivariate analysis.

In univariate analysis predicting recurrence of AP in patients with stroke and dysphagia, the risk of AP recurrence decreased by 0.2% (HR 0.998, 95% CI = 0.997–0.999, $P = 0.003$), 0.5% (HR 0.995, 95% CI = 0.992–0.999, $P = 0.015$), and 51% (HR 0.489, 95% CI = 0.277–0.861, $P = 0.013$) for increasing T2MI, Min PCF and whether tube out or not, respectively. Otherwise, the risk of AP recurrence was increased by 3.8% for each year of age (HR 1.038, 95% CI = 1.015–1.063, $P = 0.001$). However, multivariate analysis showed that the risk of AP recurrence decreased by 0.1% for increasing T12MI (adjusted HR (aHR): 0.999, 95% CI = 0.997–1.000, $P = 0.038$), but decreased by 3.4% (aHR: 1.034, 95% CI = 1.009–1.059, $P = 0.007$) for each year of age in patients with stroke and dysphagia disorder (Table 4).

**Table 1. Basic demographic characteristics of the participants with recurrence and no recurrence of AP.**

|  | No recurrence (*N* = 180) | Recurrence (*N* = 88) | *P* value | SMD |
|---|---|---|---|---|
| Sex, male | 134 (74.4%) | 62 (70.5%) | 0.586 |  |
| Age, years | 68.26 ± 13.31 | 73.68 ± 10.87 | <0.001 | 0.09 |
| BMI | 21.27 ± 3.54 | 20.20 ± 2.93 | 0.014 | -0.06 |
| T12 muscle index | 708.1 ± 229.9 | 622.3 ± 184.1 | 0.001 | -0.345 |
| Comorbidities |  |  |  |  |
| DM | 76 (42.5%) | 41 (46.6%) | 0.611 |  |
| Hypertension | 139 (77.7%) | 61 (69.3%) | 0.185 |  |
| COPD | 27 (15.2%) | 16 (18.4%) | 0.624 |  |
| CAD | 29 (16.29%) | 12 (13.64%) | 0.424 |  |
| Respiratory pressure meter |  |  |  |  |
| MIP (cmH$_2$O) | 12.2 ± 15.8 | 9.9 ± 12.2 | 0.205 | 0.23 |
| MEP (cmH$_2$O) | 20.1 ± 23.0 | 17.6 ± 17.0 | 0.305 | 0.12 |
| PCF (L/min) | 84.6 ± 91.6 | 57.8 ± 59.2 | 0.004 | 0.44 |
| MASA | 130.7 ± 40.2 | 125.5 ± 39.9 | 0.322 | 0.03 |
| GUSS | 6.8 ± 5.7 | 6.4 ± 5.1 | 0.549 | 0.04 |
| Successful NG tube weaning | 73 (40.6%) | 22 (25.0%) | 0.018 |  |
| FOIS | 1.31 ± 0.95 | 1.28 ± 0.86 | 0.857 | 0.13 |
| PAS | 7.26 ± 1.63 | 7.41 ± 1.44 | 0.457 | 0.10 |
| MBI | 18.9 ± 27.3 | 17.0 ± 24.7 | 0.570 | 0.066 |
| MMSE | 11.2 ± 10.1 | 12.2 ± 10.7 | 0.455 | 0.014 |
| Berg | 10.3 ± 16.0 | 10.4 ± 16.2 | 0.958 | 0.0013 |

Data are presented as mean ± standard deviation, or number (%).

SMD: standardized mean difference, AP: aspiration pneumonia, BMI: body mass index, DM: diabetes mellitus, COPD: chronic obstructive lung disease, CAD: coronary artery disease, MIP: maximal inspiratory pressure, MEP: maximal expiratory pressure, PCF: peak cough flow, MASA: Mann Assessment of Swallowing Ability, GUSS: gugging swallowing screen, NG: nasogastric, FOIS: functional oral intake scale, PAS: penetration aspiration scale, MBI: modified Barthel index, MMSE: mini-mental state examination

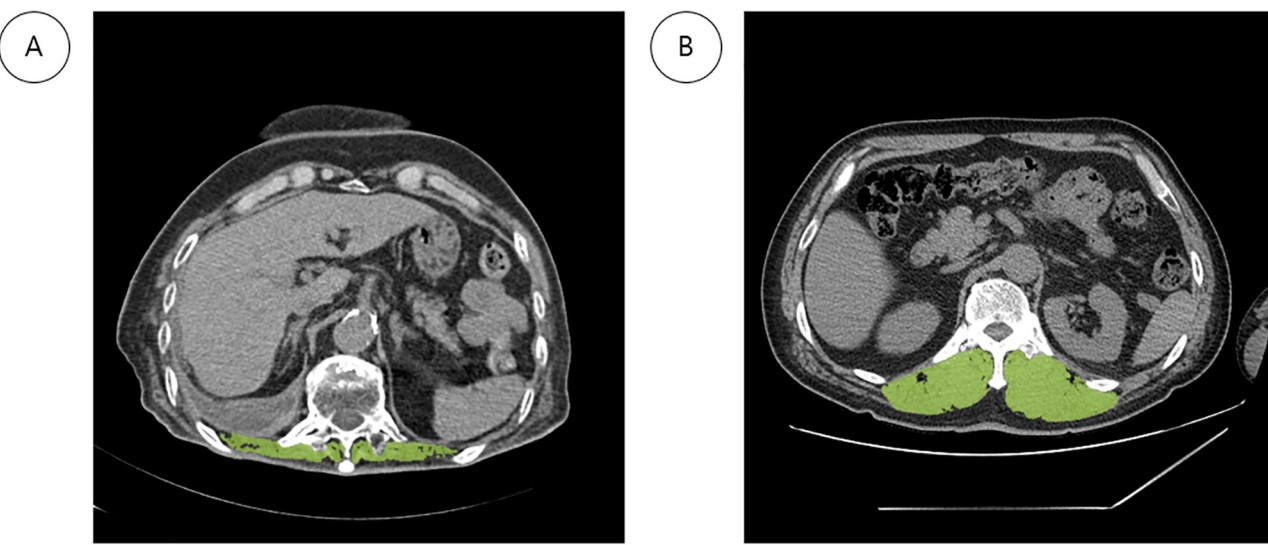

**Fig 2. Comparison of paraspinal thoracic muscles at the T12 level in aspiration.** Pneumonia with (**A**) and without (**B**) recurrence. Non-enhanced chest CT images were assessed for CSA, which was performed semi-automatically with Syngo. Via Client (ver. 5.1.0090, Siemens Healthcare GmbH Henkestr, Erlangen, Germany). CSA of ESMs in green.

**Table 2. Laboratory findings at the diagnosis of AP.**

| | No recurrence (N = 180) | Recurrence (N = 88) | P value |
|---|---|---|---|
| WBC (cells/µL, $10^3$) | 11.4 ± 4.9 | 10.3 ± 4.9 | 0.082 |
| Seg (%) | 73.7 ± 14.1 | 73.5 ± 13.2 | 0.927 |
| Lympho (%) | 16.5 ± 12.0 | 16.5 ± 10.4 | 0.976 |
| Hemoglobin | 11.9 ± 2.1 | 11.7 ± 2.1 | 0.352 |
| Platelet | 264.5 ± 98.6 | 256.8 ± 93.9 | 0.547 |
| NLR | 7.8 ± 7.5 | 7.4 ± 6.4 | 0.649 |
| PLR | 224.2 ± 180.5 | 249.9 ± 264.4 | 0.412 |
| CRP | 50.7 ± 65.3 | 43.6 ± 67.1 | 0.412 |
| BUN | 20.2 ± 15.4 | 22.5 ± 15.4 | 0.240 |
| Creatinine | 1.0 ± 0.9 | 1.2 ± 1.6 | 0.236 |
| Albumin | 3.7 ± 0.6 | 3.6 ± 0.7 | 0.354 |
| Protein | 6.7 ± 0.8 | 6.9 ± 0.9 | 0.146 |

Data are presented as mean ± standard deviation, or number (%).

AP: aspiration pneumonia, WBC: white blood cell, NLR: neutrophil-lymphocyte ratio, PLR: platelet-lymphocyte ratio, CRP: C-reactive protein, BUN: blood urea nitrogen

**Table 3. Pearson correlation coefficients (r) for the relation of T12 muscle index and other parameters in patients with AP.**

| | T12 muscle index | |
|---|---|---|
| | r | P value |
| CRP | -0.1238 | 0.043 |
| Albumin | 0.2954 | <0.001 |
| Protein | 0.1555 | <0.05 |
| Hemoglobin | 0.2798 | <0.001 |
| MIP ($cmH_2O$) | 0.2588 | <0.001 |
| MEP ($cmH_2O$) | 0.2469 | <0.001 |
| PCF (L/min) | 0.2666 | <0.001 |

AP: aspiration pneumonia, CRP: C-reactive protein, MIP: maximal inspiratory pressure, MEP: maximal expiratory pressure, PCF: peak cough flow

**Table 4. Univariate and multivariate analysis predicting recurrence of AP in patients with dysphagia.**

| Variables | Univariate analysis | | Multivariate analysis | |
|---|---|---|---|---|
| | HR (95% CI) | P value | Adjusted HR (95% CI) | P value |
| T12MI | 0.998 (0.997–0.999) | 0.003 | 0.999 (0.997–1.000) | 0.038 |
| Age | 1.038 (1.015–1.063) | 0.001 | 1.034 (1.009–1.059) | 0.007 |
| PCF ($cmH_2O$) | 0.995 (0.992–0.999) | 0.015 | 0.998 (0.994–1.002) | 0.323 |
| Albumin | 0.818 (0.534–1.251) | 0.353 | 1.196 (0.743–1.924) | 0.461 |
| Removal of NG tube | 0.489 (0.277–0.861) | 0.013 | 0.583 (0.310–1.093) | 0.092 |

AP: aspiration pneumonia, HR: hazard ratio, CI: confidence interval, PCF: peak cough flow, NG: nasogastric
[a]Multivariate analysis was executed with the following variales: T12MI, age, PCF, albumin, and removal of NG tube

## Discussion

This study identified the clinical factors associated with recurrence of AP in patients with stroke and dysphagia disorder. During the study period, AP recurred in 32.8% of all included patients. The recurrence group showed higher mean age, lower BMI and T12MI, and poorer voluntary cough strength than the non-recurrence group. T12MI was significantly correlated with laboratory parameters including albumin levels, and respiratory muscle strength. Lower T12MI and older age were independent predictors of AP recurrence in patients with stroke and dysphagia in multivariate analysis, even after adjusting for other clinical factors that increase AP risk.

Dysphagia is present in about 27–50% of patients with acute stroke, and is associated with an increased risk of AP and malnutrition [36]. To prevent these complications, accurate diagnosis of dysphagia and appropriate interventions such as behavioral strategies, enteral feeding using nasogastric tubes, or percutaneous endoscopic gastrostomy have been performed in clinical settings [37–39]. Pneumonia is one of the leading causes of mortality after a stroke and accounted for approximately 35% of poststroke deaths [40]. Most stroke-related pneumonia is dysphagia related AP [17, 18]. Compared to non-AP, AP required longer hospital stays and had higher mortality and recurrence rates [5, 41]. AP is one of the risk factor of pneumonia recurrence, and recurrent pneumonia can cause functional decline resulting in a poor prognosis for patients with AP [41]. For these reasons, specific management aimed at preventing the recurrence of AP and identifying risk factors for AP recurrence is needed.

A previous study demonstrated AP resulted in muscle atrophy involving the respiratory, skeletal and swallowing systems in a preclinical animal model and in human patients [42]. The mechanism is explained by the fact that inflammation causes the production of pro-inflammatory cytokines and induces muscle weakness through proteolysis [43, 44]. Also, loss of skeletal muscle mass can be induced by inactivity, malnutrition, and enhanced energy expenditure [45]. Moreover, repeated AP causes further muscle atrophy and the prevention of recurrence is clinically important. However, the effect of sarcopenia on recurrence of AP has not been reported.

Several modalities have been used to assess low skeletal mass index related to sarcopenia. Bioelectrical impedance analysis, dual energy X-ray absorptiometry, B-mode ultrasound, and magnetic resonance imaging are generally used to quantify skeletal muscle mass [46–48]. Measurement of the CSA of skeletal muscles on single slice axial CT scans is an alternative method to assess local skeletal muscle mass [49, 50]. Antigravity muscles reflect physical activity more than other muscle groups do, and atrophy associated with chronic bed rest was more marked in the antigravity muscles, such as the back and transversus abdominis [48]. This study focused on the ESMs, one of the major antigravity muscle groups that can be assessed using chest CT, which is routinely performed to diagnose and evaluate AP and differentiate other critical pulmonary diseases. In a previous study, $ESM_{CSA}$ assessed by chest CT was a valuable clinical parameter to correlate symptoms and disease prognosis in patients with chronic obstructive lung disease [51].

In the present study, T12MI correlated significantly with albumin, protein, and hemoglobin. Also, T12MI was negatively correlated with CRP in patients with stroke and AP. This study focused solely on measuring skeletal muscle mass and did not have access to other functional parameters, such as grip strength, which are also essential for fully satisfying the diagnostic criteria for sarcopenia. Despite this limitation, our T12MI results align with findings from previous studies that investigated clinical factors associated with sarcopenia. Serum albumin is one of the blood biomarkers representing sarcopenia. In meta-analysis, blood albumin levels were found among the elderly adults presenting with statistically significant sarcopenia

[52]. Sarcopenic patients had significantly higher inflammatory markers such as erythrocyte sedimentation rate, CRP, and lower levels of hemoglobin, albumin, and total protein [53]. Although the etiology of sarcopenia is multifactorial, chronic inflammation has been strongly implicated in muscle wasting. Inflammation is involved in the activation of many molecular pathways affecting skeletal muscle wasting, leading to an imbalance between protein synthesis and catabolism [54]. Malnutrition is also involved in increased loss of muscle mass with aging, and results in increased morbidity and mortality [55]. Similarly, T12MI was correlated with higher nutrient laboratory and lower inflammatory markers in this study.

The current study found that respiratory muscle strength as assessed by the MIP, MEP and PCF were significantly correlated with T12MI. Sarcopenia is not solely confined to peripheral skeletal muscles but overall loss of skeletal muscle mass and strength. In a previous study, MIP and MEP in elderly subjects were reported to correlate with peripheral muscle strength in elderly patients, which indicates that respiratory and peripheral muscle strength is inter-related [56]. In aging mice, sarcopenia of the diaphragm muscle has been demonstrated. Thus, it is suggested that sarcopenia may limit the ability of the diaphragm muscle to accomplish expulsive, non-ventilatory behaviors for airway clearance and contribute to respiratory complications [57].

An intriguing finding of this study is that traditional dysphagia and swallowing severity parameters associated with post-stroke pneumonia—namely, FOIS, PAS, GUSS, and MASA—no longer demonstrated significant associations with the recurrence of AP in univariate analysis. Notably, while PCF emerged as a significant predictive factor for AP recurrence in univariate analysis, it failed to reach statistical significance in multivariate analysis. Instead, age and the T12MI were identified as significant predictors of AP recurrence. This shift underscores those distinct factors, beyond traditional dysphagia assessments and PCF, play a role in predicting AP recurrence in stroke patients. The clinical relevance of T12MI is particularly compelling: while measuring respiratory muscle strength (e.g., PCF) necessitates the use of spirometry and requires patient cooperation [58], T12MI can be automatically calculated from routine chest CT scans. This provides an easily accessible and non-invasive biomarker for assessing risk in patients early on, potentially improving the identification of those at high risk for AP recurrence based on initial imaging alone.

This study had several limitations. First, the sample size was small. Second, a standard method for measuring the $ESM_{CSA}$ has not yet been established. As in a previous study, the researchers analyzed $ESM_{CSA}$ via manual shading of a specific muscle area after the application of a density mask between 40 and 90 HU. Automated programs to assess $ESM_{CSA}$ are necessary to allow a more precise and objective analysis and clinical application in patients with AP. Third, the physical activity of participating patients which may affect $ESM_{CSA}$ was not investigated because the study was performed retrospectively. The stroke occurrence time and site and nutrition states differed from patient to patient, so the degree of physical activity, including chronic bed rest time, was different. A potential limitation of this study was the handling of missing data. To handle missing entries, we employed mean imputation; missing values were replaced with the mean calculated from the available data for each respective variable. Finally, findings of the present study were confined to a single center. Future multicenter prospective studies are warranted to replicate our findings.

Prediction of any AP in patients with post-stroke dysphagia has been well reported. Those who once had AP are at repetitive risk of future respiratory events and poor clinical outcomes [59]. Nevertheless, to date, there are no guidelines on how to predict these recurrences. The T12MI can prove valuable in predicting those at risk of repetitive AP and help provide strategies that may help prevent these recurrences. Future studies are warranted to identify the threshold values that can facilitate accurate screening of these patients. Additionally, further

investigation is needed to compare the skeletal muscle mass indexes and other relevant indices in sarcopenia.

## Conclusion

Our findings demonstrate that a T12M1 at the time of first AP at the acute stroke period may be a useful predictive factor for the recurrence of AP in patients with stroke. To effectively prevent the recurrence of AP in patients with stroke and dysphagia, future studies should focus on investigating the potential role of this biomarker in guiding nutrition management and rehabilitation interventions. Such research efforts would offer valuable insights into the development of targeted strategies to mitigate the risk of AP in this vulnerable patient group.

## Supporting information

**S1 Fig. Comparison of time to recurrence (days) of aspiration pneumonia in patients with dysphagia according to T2MI quartiles.**
(DOCX)

**S1 Data.**
(XLSX)

## Author Contributions

**Conceptualization:** Sun Im, Hye Seon Kang.

**Formal analysis:** Sun Im, Hye Seon Kang.

**Investigation:** Bo Mi Gil, Yu Jin Hong.

**Methodology:** Sun Im, Hye Seon Kang.

**Writing – original draft:** Bo Mi Gil, Sun Im, Hye Seon Kang.

**Writing – review & editing:** Bo Mi Gil, Sun Im, Yu Jin Hong, Hye Seon Kang.

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
