## [Decision Letter · Decision Letter 0]

20 Aug 2024

PONE-D-23-29731Low thoracic skeletal mass index, A novel marker to predict recurrence of aspiration pneumonia in the elderly stroke patientsPLOS ONE

Dear Dr. Kang,

Thank you for submitting your manuscript to PLOS ONE. After careful consideration, we feel that it has merit but does not fully meet PLOS ONE’s publication criteria as it currently stands. Therefore, we invite you to submit a revised version of the manuscript that addresses the points raised during the review process.

We look forward to receiving your revised manuscript.

Kind regards,

Dong Keon Yon, MD, FACAAI, FAAAAI

Academic Editor

PLOS ONE

“This work was supported by a National Research Foundation of Korea (NRF) grant funded by the Korean government (no. 2020R1F1A1065814). Sun Im received this fund.”

5. We note that you have indicated that there are restrictions to data sharing for this study. PLOS only allows data to be available upon request if there are legal or ethical restrictions on sharing data publicly. For more information on unacceptable data access restrictions, please see http://journals.plos.org/plosone/s/data-availability#loc-unacceptable-data-access-restrictions.

6. In this instance it seems there may be acceptable restrictions in place that prevent the public sharing of your minimal data. However, in line with our goal of ensuring long-term data availability to all interested researchers, PLOS’ Data Policy states that authors cannot be the sole named individuals responsible for ensuring data access (http://journals.plos.org/plosone/s/data-availability#loc-acceptable-data-sharing-methods).

7. Please ensure that you include a title page within your main document. You should list all authors and all affiliations as per our author instructions and clearly indicate the corresponding author.

Additional Editor Comments:

Please address the reviewers' comments appropriately. If not, I cannot guarantee acceptance.

Reviewers' comments:

Reviewer's Responses to Questions

**Comments to the Author**

1. Is the manuscript technically sound, and do the data support the conclusions?

Reviewer #1: Yes

Reviewer #2: Yes

2. Has the statistical analysis been performed appropriately and rigorously? 

Reviewer #1: Yes

Reviewer #2: Yes

3. Have the authors made all data underlying the findings in their manuscript fully available?

Reviewer #1: No

Reviewer #2: Yes

4. Is the manuscript presented in an intelligible fashion and written in standard English?

Reviewer #1: Yes

Reviewer #2: Yes

5. Review Comments to the Author

Reviewer #1: This is a retrospective analysis of patients diagnosed with stroke who developed aspiration pneumonia and whether thoracic skeletal muscle index was a predictor for aspiration pneumonia recurrence. The authors analyzed data from 2014-2020 in a single center. Aspiration pneumonia was defined as clinical signs/symptoms of pneumonia in the dependent portions of the lung. Measured thoracic muscle volume used the CSA of the erector spine muscle at the 12th vertebral region. Those with a lower T12MI were more likely to have a recurrence based on multivariate logistic regression analysis. Overall the paper appears to demonstrate a novel association in a large cohort of patients. I had some questions related to their methodology. There were also a few instances in the paper where the authors made statements which may be debatable (such as "80% of pneumonias that require hospitalization are aspiration pneumonias). Would highly recommend that the authors double check that they are using citations appropriately and with the correct context.

Major criteria:

• The authors divided ESM by height to develop the thoracic skeletal muscle index. Did the authors consider dividing by BMI? Several authors have divided by body size (i.e. BMI) instead of height. Please provide the rationale for dividing by height and not BMI.

• In the introduction, the authors state that aspiration pneumonia is responsible for 80% of pneumonia cases that require hospitalization, however the paper that they cite demonstrated 80% for those 70 years and older. The overall prevalence of AP was 60% of hospitalized patients with CAP. Would ensure that the authors are clear what population they are referring to when using citations. Would also include other papers and include a range of incidence for aspiration pneumonia as the cause of CAP in hospitalized patients. Most citations state that 5-15% of cases of CAP are due to aspiration pneumonia, although these may include non-hospitalized patients.

• In your methods, please include the rationale and evidence for the the muscle density you use (40 to 100 HU). Many studies use a different range (-50 to +90 for instance). (PMID: 24558953)

• The authors quantified the time to recurrence of aspiration pneumonia, but I was surprised they did not do a survival analysis.

• In the results section, the authors describe Hazard ratios but it would be helpful to provide an interpretation for the reader. Such as “each drop in T12MI by XX was associated with an increase in AP recurrence by XX”

• The authors state that “For this study, we conducted a retrospective review of data for patients diagnosed with dysphagia disorder due to stroke.” -> Do we know when the stroke was diagnosed? Are these recent strokes or happened years prior?

Minor criteria:

• Methods lines 113-114, “In stroke patients with severe facial palsy and unable to perfect lip seal, the therapist aided by holding the lips around the mouthpiece minimize air leak." -> I believe the authors meant "around the mouthpiece TO minimize air leak"

• Please double check your citations. The authors state: “In a previous study, ESMCSA assessed by chest CT was a valuable clinical parameter to correlate symptoms and disease prognosis in patients with chronic obstructive lung disease.[37] “ but the citation for 37 is a basic science paper unrelated to ESM.

• In the discussion the authors state: “Most stroke-related pneumonia is dysphagia related AP” - please provide a citation here.

Reviewer #2: 1. This manuscript presents the potential of low thoracic muscle index as an effective predictive factor for the recurrence of aspiration pneumonia in patients with stroke. While the paper is well-written, I have some minor concerns.

2. Person-first language (e.g., patient with stroke not stroke patient; or person with diabetes not a diabetic) is highly recommended in academic writing.

https://www.nih.gov/nih-style-guide/person-first-destigmatizing-language

3. Discussion line 249-250 “Our results showed that those at the time of AP diagnosis all patients already” The sentence appears to be incomplete.

4. Please check the abbreviation (e.g., confidence interval [CI] and aspiration pneumonia [AP]). To ensure clarity and readability, please use abbreviations consistently once they have been introduced.

5. It would be beneficial to include some other potential confounding factors, such as the severity of stroke or the degree of dysphagia, in multivariate analysis if feasible.

6. PLOS authors have the option to publish the peer review history of their article (what does this mean?). If published, this will include your full peer review and any attached files.

Reviewer #1: No

Reviewer #2: No

---

## [Author Response · Author response to Decision Letter 0]

4 Oct 2024

Dear Editor-in-Chief, 

We would like to thank you and the reviewers of the PLOS ONE for taking the time to review our article. It is truly honorable to receive the letter of revision to enrich the work of research we’ve performed. We’d like to appreciate for the time and efforts by the editors to this paper. We have made some corrections and clarifications in the manuscript after going over the reviewers’ comments. 

The changes are summarized below:

“This work was supported by a National Research Foundation of Korea (NRF) grant funded by the Korean government (no. 2020R1F1A1065814) and Technology Innovation Program (No. 20018182) funded by the Ministry of Trade, Industry & Energy (MOTIE, Korea), 

 : As the editor suggested, we added the sentences about the role of funders, and we also additional financial disclosure additionally.

5. We note that you have indicated that there are restrictions to data sharing for this study. PLOS only allows data to be available upon request if there are legal or ethical restrictions on sharing data publicly. For more information on unacceptable data access restrictions, please see http://journals.plos.org/plosone/s/data-availability#loc-unacceptable-data-access-restrictions.

: We allow data to be available upon request. 

: We will upload our data as supporting information files.

6. In this instance it seems there may be acceptable restrictions in place that prevent the public sharing of your minimal data. However, in line with our goal of ensuring long-term data availability to all interested researchers, PLOS’ Data Policy states that authors cannot be the sole named individuals responsible for ensuring data access (http://journals.plos.org/plosone/s/data-availability#loc-acceptable-data-sharing-methods).

7. Please ensure that you include a title page within your main document. You should list all authors and all affiliations as per our author instructions and clearly indicate the corresponding author.

 : As the editor requested, we included a title page within our main document.

Additional Editor Comments:

Please address the reviewers' comments appropriately. If not, I cannot guarantee acceptance.

Reviewers' comments:

Reviewer's Responses to Questions

Comments to the Author

5. Review Comments to the Author

Reviewer #1: This is a retrospective analysis of patients diagnosed with stroke who developed aspiration pneumonia and whether thoracic skeletal muscle index was a predictor for aspiration pneumonia recurrence. The authors analyzed data from 2014-2020 in a single center. Aspiration pneumonia was defined as clinical signs/symptoms of pneumonia in the dependent portions of the lung. Measured thoracic muscle volume used the CSA of the erector spine muscle at the 12th vertebral region. Those with a lower T12MI were more likely to have a recurrence based on multivariate logistic regression analysis. Overall the paper appears to demonstrate a novel association in a large cohort of patients. 

I had some questions related to their methodology. There were also a few instances in the paper where the authors made statements which may be debatable (such as "80% of pneumonias that require hospitalization are aspiration pneumonias). Would highly recommend that the authors double check that they are using citations appropriately and with the correct context.

: Thank you for your valuable pointing that out. We corrected sentences and detailed specific population. (Line 51-54)

Major criteria:

• The authors divided ESM by height to develop the thoracic skeletal muscle index. Did the authors consider dividing by BMI? Several authors have divided by body size (i.e. BMI) instead of height. Please provide the rationale for dividing by height and not BMI.

: Thank you for your valuable comments. Sarcopenia index is different from studies. In some prediction models for sarcopenia, skeletal muscle, sex, age and body weight is included (Eur Arch Otorhinolaryngol. 2023; 280(12): 5583–5594.). We are based on previous studies using ESM and patients’ height squared, and we cited references in line 176.

• In the introduction, the authors state that aspiration pneumonia is responsible for 80% of pneumonia cases that require hospitalization, however the paper that they cite demonstrated 80% for those 70 years and older. The overall prevalence of AP was 60% of hospitalized patients with CAP. Would ensure that the authors are clear what population they are referring to when using citations. Would also include other papers and include a range of incidence for aspiration pneumonia as the cause of CAP in hospitalized patients. Most citations state that 5-15% of cases of CAP are due to aspiration pneumonia, although these may include non-hospitalized patients.

 : As the reviewer suggested, we corrected sentences and detailed specific population. (Line 54). And also added incidence of AP as the cause of CAP in hospitalized patients (Line 51-54)

• In your methods, please include the rationale and evidence for the the muscle density you use (40 to 100 HU). Many studies use a different range (-50 to +90 for instance). (PMID: 24558953)

: In previous study assess body composition, normal-density muscle is defined as having attenuation values in the 40–100 HU range. We cited related references in line 176.

• The authors quantified the time to recurrence of aspiration pneumonia, but I was surprised they did not do a survival analysis. 

: Thank you for your valuable comments. As the reviewer suggested, we did survival analysis. T2MI is continuous variables, we grouped patients into 4 groups according to T2MI quartile, but there were no differences among 4 groups in Kaplan Meier survival analysis (Log rank 0.916). We will upload as supplementary files.

• In the results section, the authors describe Hazard ratios but it would be helpful to provide an interpretation for the reader. Such as “each drop in T12MI by XX was associated with an increase in AP recurrence by XX”

: As the reviewer suggested, we changed the expressions to clarify interpretation for the reader (Line 242-251)

• The authors state that “For this study, we conducted a retrospective review of data for patients diagnosed with dysphagia disorder due to stroke.” -> Do we know when the stroke was diagnosed? Are these recent strokes or happened years prior?

: As the reviewer pointed out, we specified stroke period including acute and subacute stroke. In line 96-97 and 100-101)

Minor criteria:

• Methods lines 113-114, “In stroke patients with severe facial palsy and unable to perfect lip seal, the therapist aided by holding the lips around the mouthpiece minimize air leak." -> I believe the authors meant "around the mouthpiece TO minimize air leak"

: As the reviewer suggested, we added ‘to’ to clarify meaning in lines 138.

• Please double check your citations. The authors state: “In a previous study, ESMCSA assessed by chest CT was a valuable clinical parameter to correlate symptoms and disease prognosis in patients with chronic obstructive lung disease.[37] “ but the citation for 37 is a basic science paper unrelated to ESM.

: As the reviewer pointed out, we reviewed and cited other reference.

• In the discussion the authors state: “Most stroke-related pneumonia is dysphagia related AP” - please provide a citation here.

 : As the reviewer suggested, we cited references additionally.

Reviewer #2: 

1. This manuscript presents the potential of low thoracic muscle index as an effective predictive factor for the recurrence of aspiration pneumonia in patients with stroke. While the paper is well-written, I have some minor concerns.

2. Person-first language (e.g., patient with stroke not stroke patient; or person with diabetes not a diabetic) is highly recommended in academic writing.

https://www.nih.gov/nih-style-guide/person-first-destigmatizing-language

: Thank you for reviewer’s valuable comments. As the reviewer suggested, we corrected several phrases to person-first language.

3. Discussion line 249-250 “Our results showed that those at the time of AP diagnosis all patients already” The sentence appears to be incomplete.

: As the reviewer pointed out, we reviewed and deleted incomplete sentences.

4. Please check the abbreviation (e.g., confidence interval [CI] and aspiration pneumonia [AP]). To ensure clarity and readability, please use abbreviations consistently once they have been introduced.

: As the reviewer suggested, we rechecked the use of abbreviations and used consistently.

5. It would be beneficial to include some other potential confounding factors, such as the severity of stroke or the degree of dysphagia, in multivariate analysis if feasible.

 : We agreed the reviewer’s opinion. However, other dysphagic parameters and markers for functional disability except PCF were not significantly different between recurrence and non-recurrence groups. So, we included only min PCF for univariate and multivariate analysis.

---

## [Decision Letter · Decision Letter 1]

28 Oct 2024

PONE-D-23-29731R1Low thoracic skeletal mass index, A novel marker to predict recurrence of aspiration pneumonia in the elderly stroke patientsPLOS ONE

Dear Dr. Kang,

Thank you for submitting your manuscript to PLOS ONE. After careful consideration, we feel that it has merit but does not fully meet PLOS ONE’s publication criteria as it currently stands. Therefore, we invite you to submit a revised version of the manuscript that addresses the points raised during the review process.

We look forward to receiving your revised manuscript.

Kind regards,

Dong Keon Yon, MD, FACAAI, FAAAAI

Academic Editor

PLOS ONE

Journal Requirements:

Additional Editor Comments:

This is an excellent paper. Please see my minor comments.

#1. Considering that this study is retrospective, compliance with the Declaration of Helsinki is not required.

#2. In line 130, PCF, Maximal expiratory pressures (MEP) -> PCF, maximal expiratory pressures (MEP)

#3. Considering that SPSS is now available up to version 29, did you indeed conduct the statistical analysis using version 20?

#4. In a chart review study, it is essential to describe any missing variables that may naturally arise. This aspect appears to be missing from the manuscript.

#5. In Table 1, please describe SMD value instead of P value.

#6. What covariates were selected for adjustment in the multivariate analysis? I find it difficult to locate this information in the manuscript. Please add it to the table footnote.

Reviewers' comments:

Reviewer's Responses to Questions

**Comments to the Author**

1. If the authors have adequately addressed your comments raised in a previous round of review and you feel that this manuscript is now acceptable for publication, you may indicate that here to bypass the “Comments to the Author” section, enter your conflict of interest statement in the “Confidential to Editor” section, and submit your "Accept" recommendation.

Reviewer #2: All comments have been addressed

2. Is the manuscript technically sound, and do the data support the conclusions?

Reviewer #2: Yes

3. Has the statistical analysis been performed appropriately and rigorously? 

Reviewer #2: Yes

4. Have the authors made all data underlying the findings in their manuscript fully available?

Reviewer #2: No

5. Is the manuscript presented in an intelligible fashion and written in standard English?

Reviewer #2: Yes

6. Review Comments to the Author

Reviewer #2: Thank you for the opportunity to evaluate the revised manuscript. The authors did a good job addressing my comments.

7. PLOS authors have the option to publish the peer review history of their article (what does this mean?). If published, this will include your full peer review and any attached files.

Reviewer #2: No

---

## [Author Response · Author response to Decision Letter 1]

23 Nov 2024

Dear Editor-in-Chief, 

We would like to thank you and the reviewers of the PLOS ONE for taking the time to review our article. It is truly honorable to receive the letter of revision to enrich the work of research we’ve performed. We’d like to appreciate for the time and efforts by the editors to this paper. We have made some corrections and clarifications in the manuscript after going over the reviewers’ comments. 

The changes are summarized below:

Journal Requirements:

: As the editor requested, we did review our reference list, and there is no changes to the reference list.

Additional Editor Comments:

This is an excellent paper. Please see my minor comments.

#1. Considering that this study is retrospective, compliance with the Declaration of Helsinki is not required.

: As the editor requested, we deleted the sentence about the Declaration of Helsinki.

#2. In line 130, PCF, Maximal expiratory pressures (MEP) -> PCF, maximal expiratory pressures (MEP)

: Thank you for careful review. We corrected Maximal as maximal.

#3. Considering that SPSS is now available up to version 29, did you indeed conduct the statistical analysis using version 20?

: We checked again, and our institution is still use SPSS version 20. 

#4. In a chart review study, it is essential to describe any missing variables that may naturally arise. This aspect appears to be missing from the manuscript.

: As the editor suggested, we added sentences about the issue of missing variables in limitation (line 347-350) 

#5. In Table 1, please describe SMD value instead of P value.

: We calculated standardized mean differences for continuous variables and added values at table 1.

#6. What covariates were selected for adjustment in the multivariate analysis? I find it difficult to locate this information in the manuscript. Please add it to the table footnote.

: We added variables selected for adjustment in the multivariate analysis at the table 4 footnote.

---

## [Editor Report · Decision Letter 2]

26 Nov 2024

Low thoracic skeletal mass index, A novel marker to predict recurrence of aspiration pneumonia in the elderly stroke patients

PONE-D-23-29731R2

Dear Dr. Kang,

We’re pleased to inform you that your manuscript has been judged scientifically suitable for publication and will be formally accepted for publication once it meets all outstanding technical requirements.

Kind regards,

Dong Keon Yon, MD, FACAAI, FAAAAI

Academic Editor

PLOS ONE

Additional Editor Comments (optional):

This is an excellent paper.
---

## [Editor Report · Acceptance letter]

1 Dec 2024

PONE-D-23-29731R2 

PLOS ONE

Dear Dr. Kang, 

I'm pleased to inform you that your manuscript has been deemed suitable for publication in PLOS ONE. Congratulations! Your manuscript is now being handed over to our production team.

Kind regards, 

on behalf of

Dr. Dong Keon Yon 

Academic Editor

PLOS ONE